# Resin Flow Analysis for the Foam Core Sandwich Spoiler by Vacuum-Assisted Resin Injection Process

**DOI:** 10.3390/ma15155279

**Published:** 2022-07-30

**Authors:** Chao Yan, Yishen Li, Xia Su, Qi Liu, Yuning Wang, Kai Wu, Xiaoqing Wu

**Affiliations:** 1School of Textile Science and Engineering, Tiangong University, Tianjin 300387, China; yanc005@avic.com; 2AVIC Xi’an Aircraft Industry Group Company Ltd., Xi’an 710089, China; qlys312@163.com (Y.L.); sux003@avic.com (X.S.); liuq059@avic.com (Q.L.); 18049417315@163.com (Y.W.); wk_19960000@163.com (K.W.)

**Keywords:** foam core sandwich spoiler, VARI, resin flow, FEM, resin flow front

## Abstract

This article presents the numerical analysis and experimental investigation for the manufacturing of a foam core sandwich spoiler by vacuum-assisted resin injection (VARI) process. To find an injection scheme that guarantees both a good impregnation of the preform and a filling time compatible with the process window, the finite element model (FEM) was applied to analyze the effect of different injection schemes on the resin flow front patterns. Based on the obtained results, two optimal injection schemes are selected to form the spoiler structure. The experimental results show that the best molding quality can be achieved from the thick-end injection with a thin-end exit scheme. The comparison between simulation and experimental results shows that the overall deviation of the numerical analysis on resin flow time is 15.9%.

## 1. Introduction

Sandwich composites have a variety of applications such as aerospace, marine and transportation due to the versatility provided by high flexural rigidity, high specific weight and durability [1]. They are usually produced in the aerospace industry using prepreg-based autoclave molding and adhesive bonding technologies [2]. For the manufacturing of vehicles and next-generation airplanes, many more cost-effective molding techniques have to be developed. Among them, vacuum-assisted resin infusion (VARI) [3] is a promising technology for manufacturing sandwich composites due to its high productivity and good mechanical performance of produced composites [4].

VARI is developed on the RTM (resin transfer molding) process. The RTM process is a double-sided mold, while the VARI process uses a single-sided mold, sealed with a vacuum bag fiber-reinforced material [5]. The vacuum bag is used to exclude the air in the fiber-reinforced material. Through the flow and penetration of resin, the fiber-reinforced material could be fully infiltrated. As a result, the VARI process has a simpler mold structure, higher manufacturing costs and manufacturing efficiency compared to the RTM process [6]. It also enables the production of components with complex shapes and features [7]. However, a critical issue that arises during the VARI process of composite parts is the voids formation in their interior and on the surface [8]. These voids are formed due to potential problems during the manufacturing process, such as vacuum bag leakage, low resin permeability and low compaction pressure between layers. The presence of voids in the polymer matrix can directly influence the shear strength, interlaminar, compression and transversal tensile strengths, where the matrix-dominated mechanical properties have a greater influence on the composite behavior [9,10]. The VARI process mainly consists of two main steps, i.e., mold filling [11] and curing [12]. The mold filling step is a critical aspect of the VARI process since most of VARI’s issues relate to this process [13]. To make an acceptable composite part, the preform must be impregnated entirely with resin. This is primarily determined by the resin’s fluid dynamics [14] as it flows into the fiber reinforcement.

With the incoming requirement of a weight reduction effect in carbon and aircraft structures, spoilers have been widely adopted with a composite structure. At present, prepreg-hot-pressed technology is mainly used for manufacturing. At first, the upper and lower skins are formed separately. Then, the upper and lower skins and honeycomb core are glued together using a secondary bonding method. Thus, the manufacturing process is complicated, and the manufacturing cost is high [15]. In contrast, the VARI process could make all the components in a single molding step, resulting in the rapid and low-cost manufacturing of sandwich structures [4,16]. However, the rapid manufacture of sandwich composites by VARI raises the possibility of manufacturing defects, lowering the part’s quality and performance. Sandwich assemblies with fiber layers and a segmented core have more complicated resin flow behavior than assemblies with simply fiber layers [16]. Due to the difficulty in predicting resin flow patterns, it is considerably more challenging to perform the location analysis of the gates and vents [17]. A great deal of previous research into the VARI process has focused on fibrous laminate structures [18,19,20,21,22,23]. Kim et al. [24] proposed an analytical model to predict the permeability of shear-deformed woven fabric, and the VARTM process for a U-shaped composite structure was simulated using the commercial software PAM-RTM. Soares et al. [19] developed a standard computational fluid dynamics model to simulate the resin flow process, which takes the development of viscosity with temperature and time into consideration. However, very little work on sandwich structures using the VARI process can be found. As a matter of fact, manufacturing a sandwich structure by VARI is still challenging in the aerospace field.

Process modeling can accelerate the path from conception to prototype, thus reducing industrial costs and time [25]. In this work, numerical and experimental studies were performed to manufacture a foam core sandwich structure in composite material traditionally made with the autoclave process [26,27]. The component under investigation is a spoiler, which is a large primary structure with complex geometry. The length and width of the spoiler (Figure 1a) are 1600 mm and 300 mm, respectively. The central part of the spoiler had a foam core that was 1500 mm long, 210 mm wide and 32 mm high, which is shown in Figure 1b. An inverse engineering approach was applied to determine the permeability values needed to perform the simulation process. Filling simulations based on the finite element method were conducted to evaluate the injection schemes in terms of the impregnation level of the preform and the filling time compatible with the resin gel time. Moreover, the effect of the resin distribution medium on the resin flow pattern was analyzed. The spoiler components were manufactured using a VARI mold according to the results of the process simulations.

## 2. Materials and Methods

### 2.1. Materials and Experiments

The component under study was a spoiler, as shown in Figure 1a, and the central part of the spoiler had foam core. Transversal holes of 2 mm diameter distanced 25 mm from each other were processed in the foam core. The main material used in the spoiler part is RTM6-2 epoxy resin, and the ratio of resin (part A) to curing agent (part B) is 59.5:40.5. After the mold was cleaned up, the release agent was coated on the mold’s surface, and dry carbon fiber was placed on the mold. The carbon fiber fabric is T300, and the foam core is ROHACELL RIST 51-HT rigid foam. The resin distribution medium (RDM) employed in the upper skin is RESINFLOW-90HT. The fiber preform and foam core were covered with standard vacuum bag (DPT1000), and the whole set was sealed with standard sealant tape (GS43MR). 

After the preparation work was completed, the resin was flowed into the mold by the pressure difference produced by vacuum pump, and the fiber preform was impregnated. The resin was vacuum-defoamed at 80 °C ± 5 °C for 35 min, and the prefabricated body and mold were heated to 120 °C ± 5 °C for resin injection. Subsequently, the mold temperature was heated to 180 °C, insulated for 120 min and then cooled down to room temperature to obtain the foam sandwich spoiler products. After manufacturing of the spoiler component, the interlayer pore defects between the laminate and foam interface were tested using ultrasound A-scanning machine, and the thicknesses of the spoilers were measured using the spiral micrometer equipment.

### 2.2. Flow Analysis

Nowadays, the resin flow simulation has become one of the mainstream methods for liquid molding and process scheme development [28]. The simulation model can be used to analyze the resin inlet/outlet port position and the influence of resin distribution medium (RDM) in the resin flow process. For the above-mentioned spoiler part, commercial PAM-RTM 2014 software by ESI Company (Paris, France) was used to simulate the resin flow process in this study. The simulation model consists of upper and lower skin prefabricated body, foam core and RDM. The foam sandwich spoilers are all fully layered, with 4 layers of upper skin (i.e., [0/45/45/0]) and 3 layers of lower skin (i.e., [0/45/0]), as shown in Figure 1b. Foam core is sandwiched between the upper and lower skins, as shown in Figure 1c. In order to reduce the complexity of modeling and improve the efficiency of simulation calculation, the influence of auxiliary materials such as FRP columns on the resin flow process is ignored here, as shown in Figure 1b. The foam core is treated as a rigid body, and the holes are taken as through holes. 

The following assumptions referring to the previous investigations [29,30] are used to improve the computational efficiency: (1) The resin is an incompressible fluid, and the resin density remains constant in the filling process. (2) Capillarity and inertia effects of the resin flow are ignored. (3) The preform and foam are rigid, and no deformation occurs during the filling process. The boundary conditions are set as follows:(1)At the injection port: when the constant pressure injection is applied, the injection pressure is specified as the setting value: *P* = *P*_0_. Especially for the vacuum injection, the injection pressure is set to atmospheric pressure, while the outlet pressure is set to 0, given that the flow was driven by vacuum pressure. When the constant velocity injection is applied, the injection velocity is also set to the setting value *V* = *V*_0_.(2)At the flow front, vacuum pressure is used (*P* = 0).(3)At the mold wall, velocities normal to the mold walls are zero and the pressure gradient is zero (*∂P* = *∂n* = 0).

Meanwhile, the numerical analysis is assumed to be isothermal in order to further improve the computational efficiency [30,31], which in our simulations took approximately 110–140 s of CPU (Intel(R) Xeon(R) CPU E5-1620 v3@3.50GHz (8 cores)) time. The total number of triangular elements used in the simulation model is 39,888. A sensitive mesh analysis performed on the resin flow process indicated that, with this number of elements, the simulation converges to constant values in terms of resin flow time. Increasing the number of elements does not improve the computational accuracy but only the computational time.

After the completion of the prefabricated system, as shown in Figure 1c, the resin was subsequently injected. Based on the principle of the shortest resin flow path for liquid molding, and because of the local fiber deformation caused by the injection piping, the resin flow path can only be set in the margin area. In order to find a suitable molding process solution for spoilers, herein, this thesis analyzes and investigates the effects of two injection modes and four resin distribution media on resin filling time, flow front and resin pressure, as shown in Figure 2. The two injection schemes are Case 1 and Case 2, respectively.

Case 1: resin is injected from the thicker side of the spoiler component and flows out from the thinner side.

Case 2: resin is injected from the thinner side of the spoiler component and flows out from the thicker side.

The area covered by the resin distribution medium is from the resin injection port to the foam core at different positions, such as the 1/4 area meaning that the area from the resin inlet port to the spoiler length 1/4 position is fully covered by the RDM, as shown in Figure 1b and Figure 2. The resin inlet port is a line resin injection, and three-point resin outlets are evenly distributed at the resin outlet. The fiber volume fraction was set as 51% ± 3%, and the vacuum pressure was −0.95 bar. 

### 2.3. Permeability Measurement

Due to the lack of generalized models, the permeability of fabrics and distribution media has been measured experimentally. The following equation can be obtained by integrating Darcy’s law:(1)$$l^{2}=\frac{2K\Delta P}{\mu \varphi }t$$
where *l* is the position of the flow front at time *t*, and φ is the porosity of the fabric preform. The permeability can be obtained by recording *l*^2^ − *t* of the resin flow and fitting the slope of the line:(2)$$K=\frac{a\mu \varphi }{2\Delta P}$$
where *a* is the slope of the line, and *P* is 0.1 MPa when the VARI process was used for testing. Figure 3 shows the schematic representation of the experimental setup for permeability measurement. During the experiment, a camera was used to record the position of the flow front at different moments. Meanwhile, the permeability was calculated by substituting the data into Equations (1) and (2).

As shown in Figure 4, in order to record the flow front effectively, a transparent vacuum bag was applied as the top mold, and the flow front in the experiment is almost a straight line, which is beneficial to record the location of the flow front accurately. Concerning the permeability along the thickness direction of fabric preforms, transparent tempered glass was used as the bottom mold. The time for the resin to reach the bottom mold was recorded in the experiment. Afterward, the recorded data and thickness of the preforms were substituted into Equations (1) and (2) to calculate the permeability in the thickness direction. According to the experimental measurements, the permeability of the resin distribution medium and the fiber fabric in 0° direction and 90° direction was 1.2 × 10^−9^ m^2^, 3.28 × 10^−10^ m^2^ and 2.46 × 10^−10^ m^2^, respectively. The permeability of the pores in the foam core was 4.25 × 10^−7^ m^2^. 

## 3. Results and Discussion

### 3.1. Resin Flow Time

The prediction of resin flow time for long and complex parts produced using the method of resin infusion is of prominent importance [32]. Figure 5 shows the variation of resin flow and mold filling time for different resin inlet and outlet positions as well as the area of RDM. As shown in Figure 5, the resin flow filling time in Case 1 is shorter than that in Case 2 for the same area of RDM. In Case 1, the mold filling time was reduced by 17.8%, 35.2% and 36.7% for each additional 1/4 of the area of RDM compared to 1/4 area of RDM used. Therefore, the method of using RDM can significantly improve the mold-filling efficiency of resin [33]. In Case 2, the corresponding filling times were reduced by 10.5%, 16.2% and 2.0%, respectively, with less variation in time compared to the filling time in Case 1. However, the resin flow time was longer in the 4/4 RDM than that in the 3/4 RDM. This is because the resin flow rate of the upper skin is faster when the 4/4 RDM is laid down, and after the resin flow filling of the upper skin, the resin will continue to flow to the lower skin, resulting in the reverse resin wrapping of the lower skin, so the overall resin filling time in this case is longer than that in the case of laying down 2/4 and 3/4 RDM. From the analysis of injection time, increasing the area of RDM can effectively shorten the resin flow time. Additionally, the resin flows more efficiently in Case 1 than in Case 2.

### 3.2. Resin Flow Front

The flow front during the resin filling process is an important criterion to judge the difficulty of process control [33]. Figure 6 shows the cross sections in Case 1 and Case 2. T1 and T2 are the corresponding spoiler cross-section positions, where the point of T1 is near the resin inlet position, and the point of T2 is near the outlet position. The simulation model was used to simulate the injection time of the resin flow front and the time difference between the upper and lower skins at the T1 and T2 positions for the two cases, and the results are shown in Table 1.

From the simulation results in Table 1, it can be seen that the curvature of the upper skin near the inlet port in Case 2 changes more slowly, and the resin flow distance increases compared to that in Case 1, so the diversion effect of the RDM in Case 2 is more obvious before the resin flows to the T1 position. Results also show that the resin flow front of the upper skin at the T1 position is ahead of the lower skin. The resin flow time of upper and lower skins gradually decreases with the increase of the area of RDM. The upper skin has a shorter resin flow time due to the efficient diversion effect of the RDM. In the lower skin, after the area of RDM is increased, the resin will advance rapidly in the upper skin along the RDM and then penetrate to the lower skin through the through hole of the foam core, so the flow time of the lower skin will also be shortened. The maximum difference of resin flow time between upper and lower skins is 73 s. While in the T2 position, when the area of RDM is 1/4 and 2/4, the flow front of upper skins lags behind lower skins by 26 s and 20 s, after the deceleration effect in the area without RDM. When the area of RDM increases to 3/4 and 4/4, the flow front of the upper skins is ahead of lower skins, and the maximum time difference is 81 s.

In Case 1, with the increase of RDM area, the resin flow time of upper and lower skins in the T1 position remains basically the same, and the time difference of resin flow front is 0 s. In the position near the resin inlet, the curvature changes drastically, and the distance of resin flow in the upper skins increases obviously. However, due to RDM employed in the upper skin, the resin flow rate at this location is much higher than that in the lower skin without RDM, and the final result is to ensure that the resin flows at the upper and lower skins reach the unity of the flow front at the T1 location. As the resin continues to flow to the T2 position, the resin flow time in the upper skin decreases further with the increased use area of RDM. Due to the high permeability of the RDM, the resin flow time of upper skin is reduced more. In contrast, the resin flow in the lower skin is dominated by resin permeation along the foam core through holes, and the permeation efficiency is lower than the effect of the RDM.

Overall, the resin flow gradually changes from the lower skin lead to the upper skin lead as the area of RDM increases. However, at 4/4 RDM area, the resin flow time in the lower skin under the T2 position is longer than that of 3/4 RDM. This is mainly because, under the 4/4 RDM area, the resin has completed the upper skin filling at a faster speed, and the excess resin will further flow from the upper skin to the lower skin, resulting in the reverse flow of resin, as shown in Figure 7. This situation leads to the wrapping phenomenon at the resin flow front, causing the resin filling time to be extended and the upper skin leading the lower skin flow front to 41 s.

In order to further study the resin flow process of the foam sandwich spoiler, the resin flow front changes at the T2 position, when the RDM spreads 3/4 and 4/4 areas under the two cases, are compared, respectively. As shown in Figure 8, it can be seen that in Case 2 the time difference between the upper and lower skins reaches −39 s when the area of RDM is 3/4 covered, and the phenomenon of resin flow-front wrapping does not occur. When the area of RDM is 4/4, the time difference is −60 s, and the phenomenon of resin flow-front wrapping is more obvious. Similarly, in Case 1, when the RDM area is less than 3/4, the resin has no flow-front wrapping phenomenon (as shown in Figure 9a), while when the RDM area is 4/4, there is an obvious flow-front wrapping phenomenon (as shown in Figure 9b). Therefore, both cases are very likely to produce flow-front wrapping at the resin outlet location when the area of RDM is up to 4/4, thus inducing molding defects.

### 3.3. Resin Pressure

Previous studies have already indicated that the internal mold pressure increased gradually with the flow-front advancement until the end of filling [34]. Moreover, the maximum pressure is an important parameter to be considered [35]. The variation of resin pressure at the T1 and T2 positions of the upper and lower skins in the two different cases is shown in Figure 10. Figure 10a,b shows the pressure variation at T1 and T2 of the lower skin in Case 1. Results show that the pressure at the T1 position first increases rapidly to 0.6 bar with the increase of RDM area and then increases slowly to 0.95 bar, while the pressure at the T2 position increases much more slowly than that at the T1 position. The final resin pressure reaches its maximum at 3/4 RDM area, which is close to 0.8 bar, and minimum at 1/4 RDM area, which is about 0.68 bar. Figure 10c,d shows the pressure variation at the T1 and T2 positions of the lower skin in Case 2. Overall, the change of resin pressure is much slower than that in Case 1. The final resin pressure at the T1 and T2 positions decreased in the same RDM area, with the maximum pressure at the T1 position being 0.6 bar and the minimum pressure being 0.28 bar, while the maximum pressure at the T2 position decreased more significantly, with the maximum pressure being 0.26 bar and the minimum pressure being 0.07 bar. As expected, the relatively high pressure during the resin flow process is effective in suppressing void in the sense that vacuum bagging involves actual removal of bubbles within the resin [34,35]. Therefore, the probability of defects raised during the resin flow process in Case 1 is lower than that in Case 2.

In Case 1, the T1 and T2 positions are relatively close to the resin inlet port, and the loss of resin pressure along the spoiler length is smaller, which is more conducive to the maintenance of resin pressure. Meanwhile, in Case 2, the distance of resin along the structure slope to the T1 position is longer, and the resin pressure is gradually lost along the length direction, resulting in lower pressure at the T1 position, and this phenomenon lasts until the T2 position, which affects the final structure quality. Therefore, Case 1, in which resin is injected from the thicker side of the spoiler component, is more suitable for improving the manufacturing quality of spoilers. Based on these analyses, the injection scheme of the simulation Case 1 was adopted to fabricate the spoiler.

### 3.4. Experimental Results

Figure 11 shows the test part of the spoiler manufactured by the two different cases. Based on the above simulation results, 3/4 area of the RDM area was placed in both two cases. The actual resin flow time in Case 1 was 507 s, which deviated from the simulation analysis by 10.9%, and the actual resin flow time in Case 2 was 1105 s, which deviated from the simulation analysis by about −15.9%. It can be seen that the simulation model proposed in this paper can predict the resin flow process of a spoiler structure more accurately.

#### 3.4.1. Analysis of the Injection Quality

For Case 1, the quality of both the upper and lower skins of the spoiler is good, without missing resin, and the aluminum mesh on the surface of the lower skin is flat, as shown in Figure 12a. The non-destructive testing results show that there are no obvious defects in the core area and lamination area, and the porosity is less than 1.5%, which meets the design requirements. Figure 12b shows the lower skin of the spoiler manufactured using the scheme in Case 2, from which it can be seen that in the transition area between the foam core surrounding and the skin, the surface aluminum mesh resin penetration is not sufficient and there are more folds, especially at the T1 and T2 positions. When testing with ultrasound A-scanning, it was found that there were interlayer pore defects between the laminate and foam interface, as shown in Figure 13a. Moreover, there is an obvious lack of resin phenomenon in the foam core and laminate overlap area, as shown in Figure 13b. As the above analysis shows, when Case 2 is adopted, although there is no flow-front wrapping phenomenon in the process of resin flow when 3/4 RDM area is employed, the final resin pressure at the T1 and T2 positions is reduced too much compared with that in Case 1. The lowest pressure is only 0.26 bar, which leads to more intensive porosity inside the structure and eventually has a greater impact on the internal quality of the product.

#### 3.4.2. Analysis of the Thickness

Spiral micrometer equipment was used to measure the thicknesses of the spoilers at different locations under the two cases, which were then compared with the thicknesses in the theoretical design. The thickness is measured in the position of the test piece shown in Figure 14a, and the thickness distribution measurement results are shown in Figure 14b. It can be seen that the thickness at the resin inlet area is greater than the thickness at the resin outlet area for both cases, and the thickness near the resin inlet area is thicker than the theoretical thickness. The thickness near the resin outlet location is significantly lower (in Figure 14a, points 1, 9 and 16 are the locations of the resin outlet ports). In the middle position of the spoiler (i.e., H area, as shown in Figure 14a), which is far from the location of the resin inlet and outlet ports, the thickness tends to be the average thickness. The design dimensions require the spoiler thickness to be within ±8%, and it can be seen that the thickness of the spoiler manufactured under both schemes meets the design requirements.

## 4. Conclusions

The investigation deals with a problem very popular in engineering when manufacturing processes are based on VARI. Trial and error is still today the most commonly used method for the definition of resin flow strategies. However, the interest towards the application of FE analysis on processing optimization attracts growing attention. This paper takes a foam sandwich spoiler structure as the research object, investigates the influence of different resin injection schemes and RDM areas on resin flow behavior and successfully manufactures the test parts of a spoiler that meet the design requirements by the VARI process. The main conclusions of this work can be summarized as follows:(1)By simulating the resin flow process under different process schemes of spoiler structure, the resin flow situation under different combinations of resin injection schemes and RDM areas is obtained, and the time difference of the resin flow front, pressure field distribution and flow front change of upper and lower skins of the spoiler are further analyzed. The experimental results show that the simulation model established in this paper is highly accurate.(2)Two schemes of thick-end resin injection and thin-end resin injection were tested and verified, and the results showed that the resin flow time was shorter, and the internal quality of the prepared spoiler test parts was better with the thick-end resin injection scheme. In addition, the cohesive quality between the aluminum net and composite skin is better compared to the thin-end resin injection scheme.(3)For the foam sandwich structure, there is a more obvious position dependence of the thickness when the conventional VARI process is used for molding. The closer to the injection port, the thicker the product thickness is, while the thickness tends to be average at the middle of the injection and discharge port positions.

## Figures and Tables

**Figure 1 materials-15-05279-f001:**
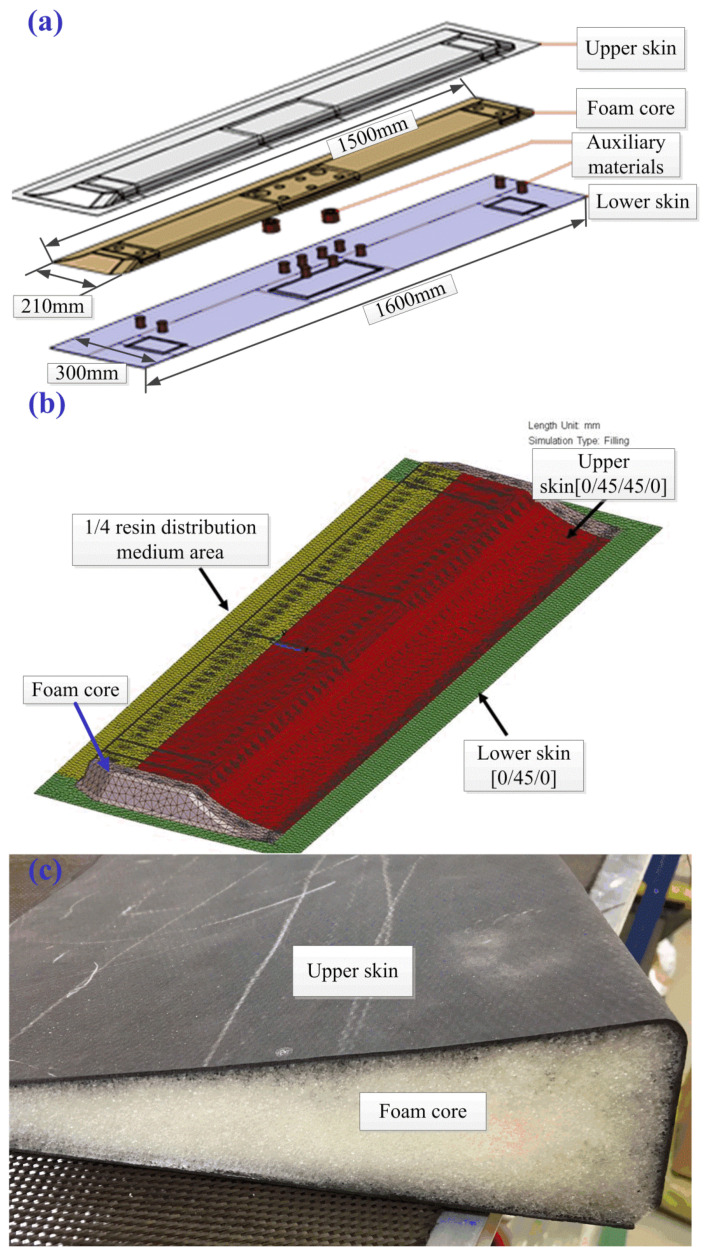
Schematic diagram of typical foam sandwich spoiler structure. (**a**) Spoiler structure; (**b**) simulation model; (**c**) spoiler prefabricated body.

**Figure 2 materials-15-05279-f002:**
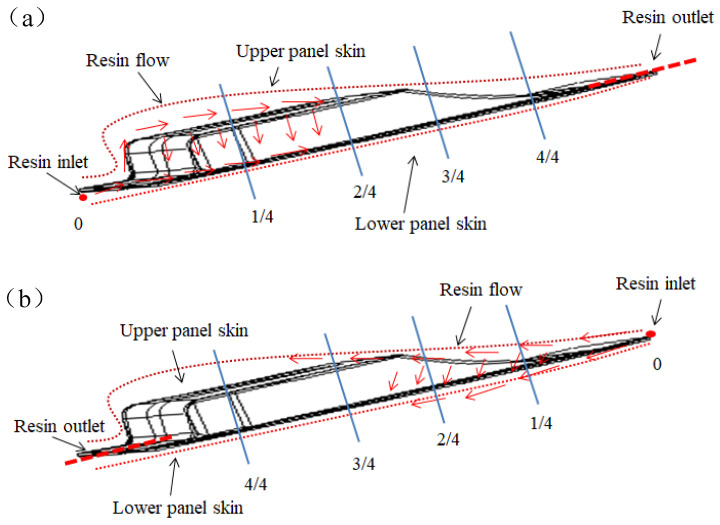
Representation of different resin inlet port on the spoiler. (**a**) Case 1; (**b**) Case 2.

**Figure 3 materials-15-05279-f003:**
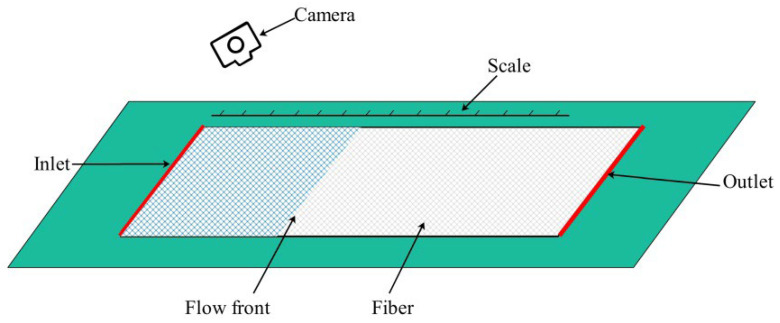
Schematic of experimental setup for the permeability measurement.

**Figure 4 materials-15-05279-f004:**
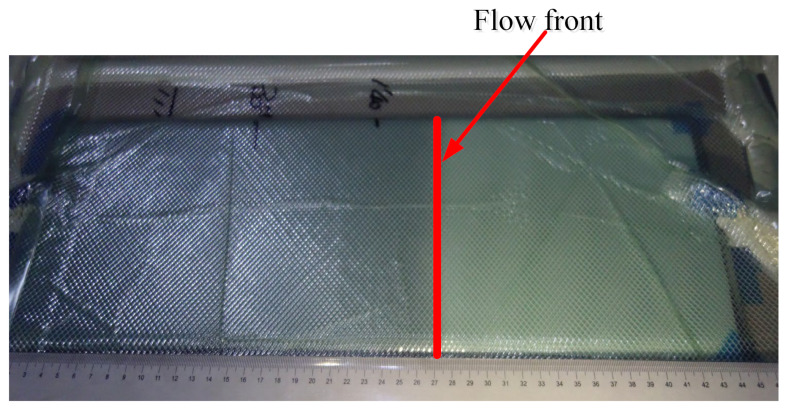
Resin flow front for the permeability measurement.

**Figure 5 materials-15-05279-f005:**
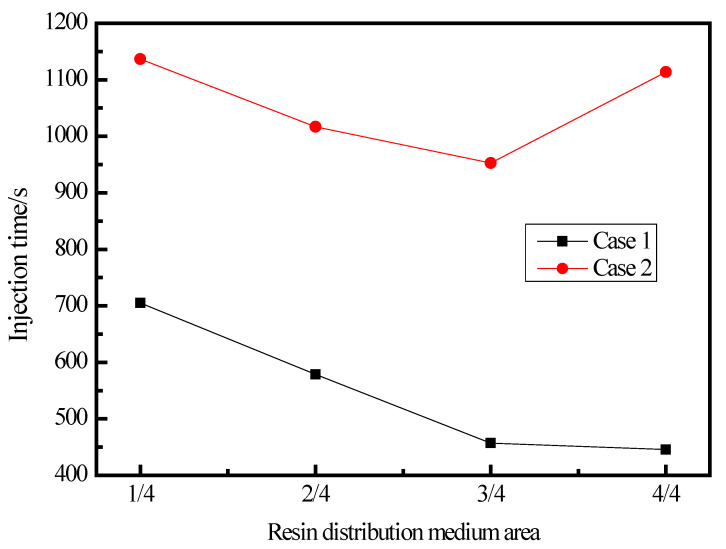
Resin flow time under different injection schemes.

**Figure 6 materials-15-05279-f006:**
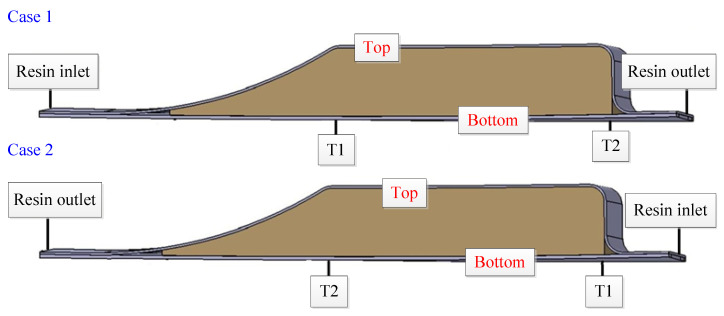
Resin flow front position under different injection cases.

**Figure 7 materials-15-05279-f007:**
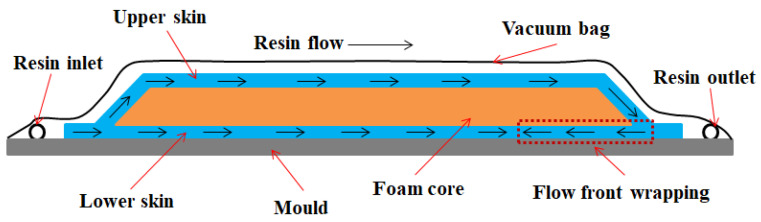
The reverse flow of resin in the lower skin under the 4/4 RDM area.

**Figure 8 materials-15-05279-f008:**
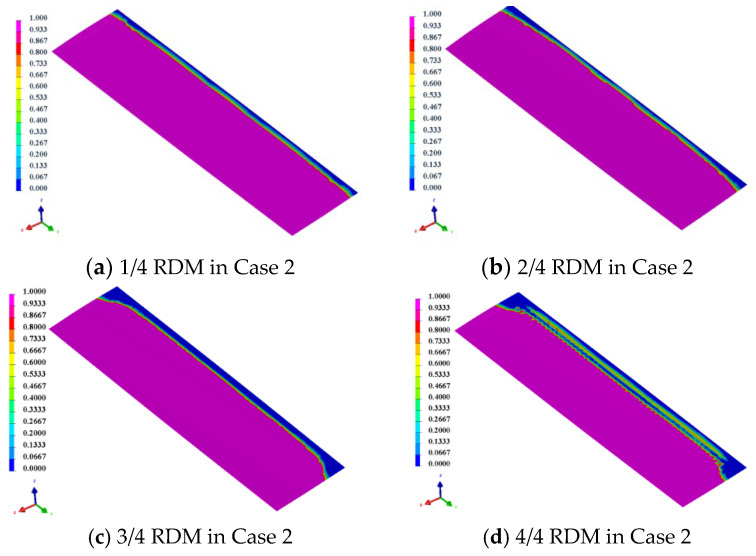
Resin flow front in Case 2.

**Figure 9 materials-15-05279-f009:**
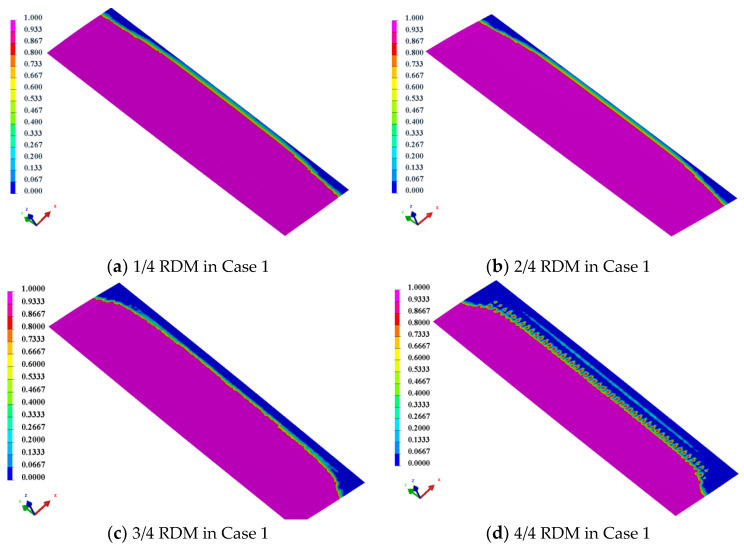
Resin flow front in Case 1.

**Figure 10 materials-15-05279-f010:**
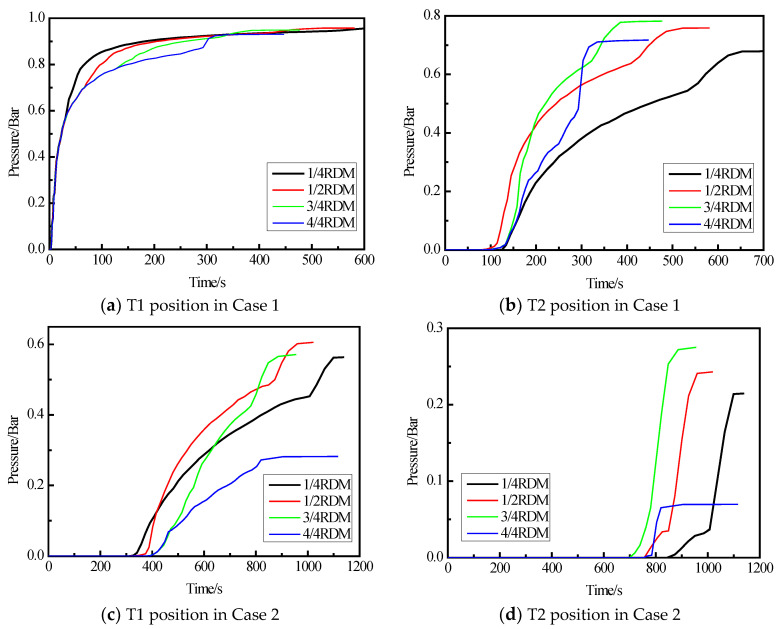
Pressure changes during resin flow under Case 1 and Case 2.

**Figure 11 materials-15-05279-f011:**
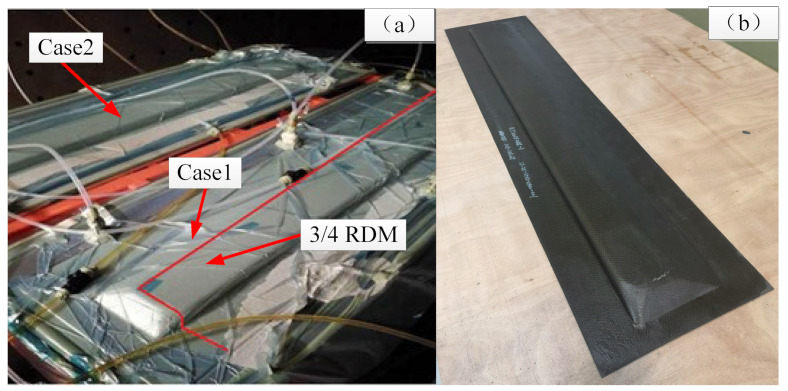
Manufacturing process of the spoiler. (**a**) Manufacturing process; (**b**) spoiler parts.

**Figure 12 materials-15-05279-f012:**
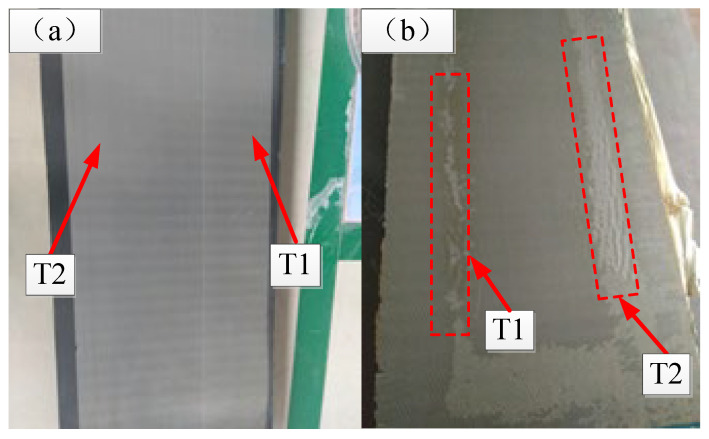
Experimental results of lower panel skin in different cases. (**a**) Case 1; (**b**) Case 2.

**Figure 13 materials-15-05279-f013:**
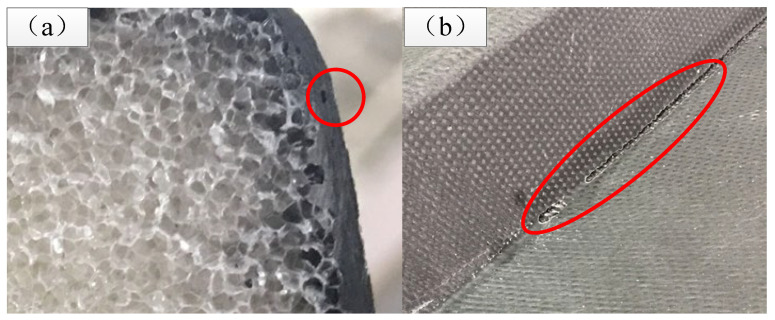
Manufacturing defects in Case 2. (**a**) Interlayer pores; (**b**) lack of resin in surface.

**Figure 14 materials-15-05279-f014:**
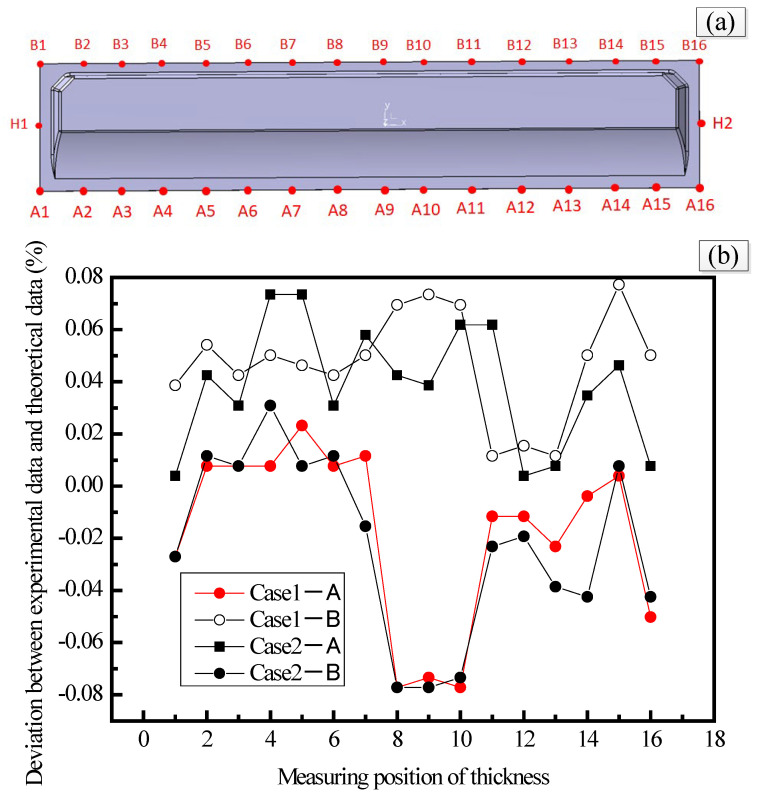
Thickness distribution of the spoiler under different cases. (**a**) Thickness measurement position; (**b**) thickness measurement results.

**Table 1 materials-15-05279-t001:** Comparison of flow front times at different positions of spoilers.

RDM Area		T1				T2			
		1/4	2/4	3/4	4/4	1/4	2/4	3/4	4/4
Case 1	Upper skin	5.1 s	5.1 s	5.2 s	5.7 s	153 s	101 s	91 s	86 s
	Lower skin	5.1 s	5.1 s	5.2 s	5.7 s	143 s	121 s	116 s	127 s
	Time difference	0 s	0 s	0 s	0 s	10 s	−20 s	−25 s	−41 s
Case 2	Upper skin	296 s	264 s	257 s	256 s	950 s	825 s	739 s	672 s
	Lower skin	341 s	335 s	329 s	329 s	924 s	805 s	763 s	753 s
	Time difference	−45 s	−71 s	−72 s	−73 s	26 s	20 s	−24 s	−81 s

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
