# Peer review of "Resin Flow Analysis for the Foam Core Sandwich Spoiler by Vacuum-Assisted Resin Injection Process"

_materials, 2022, doi:10.3390/ma15155279_

Round 1
Reviewer 1 Report
In this research, authors show out the numerical analysis and experimental of the vacuum-assisted resin injection (VARI) process. In general, authors have to check these issues:
- Line 17 – 19: Which result is mentioned?
- Authors should add more keywords
- In the Introduction part, the VARTM (Vacuum assisted resin transfer molding) and VARI should be mention and compared
- In the Introduction part: the advantage and disadvantage of VARI should be shown out. The reference is also needed more for this information
- Line 32: Reference [6] just mention about the injection molding for thin wall, so, the reference about the curing should be added
- The Figure of the product size should be added
- The description of product should be added in this research (size, function, why it was selected for this research, the different issues in manufacturing,…)
- Figure 2: the “1/4; 2/4; 3/4; 4/4” need to be described. The location of these section should be made clear
- The process parameters need to be listed (in a table) for both cases
- The reason for selected process as in case 1 and case 2 should be mentioned
- Figure 3: the format of unit of injection time need to be checked
- Figure 4: which case is at the top or bottom? Authors have to add the location of T1 and T2
- Table 1: there are 2 line with the name as “Upper skin:, so, authors have to check this issue
- Figure 5 and 6: the number is too small and it is not clear
- Figure 8: The picture should cover all the experiment model. This picture should have some notes
- Line 246 – 248: where is the thickness value of H?
- Figure 10: “Deviation between…” need the unit
Reviewer 2 Report
The paper explores the VARI process simulation using PAM-RTM followed by manufacturing the spoiler component under study. The subject at hand is of interest to the advanced composite community, and the paper is well-written in English. Before the article can be published, some issues must be further elaborated on and improved.
Page 1, line 37: authors claim that:
“By integrating all components in a single molding process, one can reduce manufacturing costs and increase production rates.”
There is no reference to back this up, and there is no explanation of what they actually mean by “a single molding process.” Perhaps they should provide more detail and references.
Numerous works on the analysis of the VARI method have been performed over the past few decades, both on simulation and experimental study of this process. Referring to only 17 studies is low, and more literature survey is required.
The authors claim that the samples were manufactured according to the process simulation results, but it is unclear what recommendations were given for the manufacturing?
Section 2.2: the laminate code on page 3, line 82, is not consistent with figure 1, or perhaps the text is not well written.
The model built in PAM-RTM is not well presented. Providing more details on boundary conditions, FE mesh, applied loads, and the type of element used is necessary. With the total number of elements around 40000, the mesh seems coarse. There is also no comparison of the simulation CPU time and its efficiency.
The presented permeability values are mysterious, and there is no explanation of where or how they are obtained.
In VARI, the permeability of the fabric is deformation dependent. Have authors considered how much deformation was invoked to the configuration during the process due to the vacuum application and the consequent resin pressure built up? This issue needs to be addressed both in experiment and simulation.
In the VARI, unlike RTM, the underlying configuration will be vacuumed and consequently changes thickness. Has been any effect seen from this? The foam core is considered rigid in the study, but thickness analysis was performed during the experiments. How is this affecting the simulation?
In general, section 2 of the paper is vague and requires further restructuring and extension of the presented details.
Table 1 has typos. The items should be Upper and lower skin, and it is not clear which one is which! Make sure that the table is placed on one page.
The legends in Figures 5, 6, and 7 are very blurred and hard to read.
Figures 5 and 6 should be improved by giving the flow front for all four sections to see the behavior in general.
Is the flow front monitored during experiments? Is the behavior similar to the one obtained in the simulation?
The simulation should be used to optimize the experiments and prevent further trial and error. Although some remarks are made here, and the authors claim that their model is highly accurate, there is no modeling strategy provided, and no recommendations are made.
The paper requires further improvements before I can recommend it for publication.
Reviewer 3 Report
This study investigated the vacuum-assisted resin injection process for the foam core sandwich spoiler using numerical and experimental approaches. The authors concluded that the best molding quality can be achieved from the thick-end injection with thin-end exit scheme. Additionally, the overall deviation between the simulation and experimental results is less than 20%.
General comment:
This study is important and quite interesting. Unfortunately, the manuscript is not well prepared. The detailed description for the setup of the experimental tests is unclear. Additionally, the results and findings are not fully discussed. In fact, there is no citation in the Discussion section. This means that the authors didn’t discuss their results and findings with any past studies.
Specific comments:
Comment 1:
Lines 78-79, “For the above-mentioned spoiler part, PAM-RTM 2014 software was used to simulate the resin flow process in this thesis.”
The detailed information for PAM-RTM 2014 software is needed to be provided, such as company, city, and country.
It is recommended to use “in this study” instead of “in this thesis”.
Comment 2:
Lines 86-87, “. The total number of elements used in the simulation model is 39888.”
How to determine the total number of elements? Do you conduct a convergence study? Please provide the result of the convergence study.
Comment 3:
The authors didn't discuss or compare their findings with previous studies in the Discussion section. In fact, there is no citation in the Discussion section. Please improve the Discussion section according to the suggestions of this comment.
Comment 4:
In Table 1, why each case has two upper skin results? Is there any mistake in this table?
Comment 5:
Lines 170-172, “This is mainly because under the 4/4 RDM area, the resin has completed the upper skin filling at a faster speed, and the excess resin will further flow from the upper skin to the lower skin, resulting in the reverse flow of resin.”
A new figure is suggested to demonstrate this phenomenon.
Comment 6:
Figure 6 was not cited in the manuscript. In fact, Figure 1a was also not cited in the manuscript.
Comment 7:
The resolution of Figure 9 is not good. It is not clear to understand the difference between the case 2 and case 1.
Comment 8:
Lines 239-241, “Spiral micrometer equipment was used to measure the thicknesses of the spoilers at different locations under two Cases, which was then compared with the thicknesses in the theoretical design.”
I think that this is a design of the study. These sentences should be mentioned in the Materials and Methods section. It is recommended that a new section, which describes the experimental setups and tests of a foam core sandwich spoiler by vacuum-assisted resin injection, is suggested to add in the manuscript.
Comment 9:
It was suggested to condense the Conclusion section. A lot of sentences in this section belong to the introduction section and materials and methods section. Please rewrite the Conclusion section.
Round 2
Reviewer 1 Report
Dear authors,
The new version was improved. However, for publishing, this paper should have some minor modification as:
1. In the Abstract: line 18 has the different percent number (between the first and the second version), so, author should check again.
2. Keywords: (line 19) “Flow front” should be “resin flow front”
3. Figure 1c and 1b should be separated
4. Figure 3 (old version) and Figure 5 (new version) have no change, please check again.
5. Figure 4 (old) and 6 (new) should have the note about case 1 and case 2:
5. The paper should be formatted

Reviewer 3 Report
I have only one suggestion for this article. The section "4. Experimental results" can be merged into the section "3. Results and discussion".
